# Effectiveness and Cost-Effectiveness of Self-Management Interventions for Adults Living with Heart Failure to Improve Patient-Important Outcomes: An Evidence Map of Randomized Controlled Trials

**DOI:** 10.3390/healthcare12030302

**Published:** 2024-01-24

**Authors:** Marilina Santero, Yang Song, Jessica Beltran, Melixa Medina-Aedo, Carlos Canelo-Aybar, Claudia Valli, Claudio Rocha, Montserrat León-García, Karla Salas-Gama, Chrysoula Kaloteraki, Ena Niño de Guzmán, Marta Ballester, Ana Isabel González-González, Rune Poortvliet, Marieke van der Gaag, Cristina Spoiala, Pema Gurung, Fabienne Willemen, Iza Cools, Julia Bleeker, Angelina Kancheva, Julia Ertl, Tajda Laure, Ivana Kancheva, Kevin Pacheco-Barrios, Jessica Hanae Zafra-Tanaka, Sofia Tsokani, Areti Angeliki Veroniki, Georgios Seitidis, Christos Christogiannis, Katerina Maria Kontouli, Oliver Groene, Rosa Sunol, Carola Orrego, Monique Heijmans, Pablo Alonso-Coello

**Affiliations:** 1Iberoamerican Cochrane Centre, Biomedical Research Institute Sant Pau (IIB Sant Pau), 08025 Barcelona, Spain; msantero@santpau.cat (M.S.); mmedinaa@santpau.cat (M.M.-A.); ccanelo@santpau.cat (C.C.-A.); cvalli@fadq.org (C.V.); monselg8@gmail.com (M.L.-G.);; 2Avedis Donabedian Research Institute (FAD), 08037 Barcelona, Spainrsunol@fadq.org (R.S.);; 3Quality, Process and Innovation Direction, Vall d’Hebron Hospital Universitari, 08035 Barcelona, Spain; 4Network for Research on Chronicity, Primary Care, and Health Promotion (RICAPPS), 08007 Barcelona, Spain; 5Netherlands Institute for Health Services Research (NIVEL), 3513 Utrecht, The Netherlands; runepoortvliet@gmail.com (R.P.); 2832547k@student.gla.ac.uk (A.K.); i.k.kancheva@lumc.nl (I.K.); m.heijmans@nivel.nl (M.H.); 6Faculty of Medicine, Universitat Autònoma de Barcelona (UAB), 08025 Barcelona, Spain; 7Department of Primary Education, School of Education, University of Ioannina, 45110 Ioannina, Greece; 8Knowledge Translation Program, Li Ka Shing Knowledge Institute, St. Michael’s Hospital, Toronto, ON M5B 1W8, Canada; areti-angeliki.veroniki@unityhealth.to; 9Institute for Health Policy, Management, and Evaluation, University of Toronto, Toronto, ON M5S 3G8, Canada; 10OptiMedis, 20095 Hamburg, Germany; 11Faculty of Management, Economics and Society, University of Witten/Herdecke, 58455 Witten, Germany

**Keywords:** heart failure, self-management interventions, randomized controlled trials, systematic review

## Abstract

Self-management interventions (SMIs) may enhance heart failure (HF) outcomes and address challenges associated with disease management. This study aims to review randomized evidence and identify knowledge gaps in SMIs for adult HF patients. Within the COMPAR-EU project, from 2010 to 2018, we conducted searches in the databases MEDLINE, CINAHL, Embase, Cochrane, and PsycINFO. We performed a descriptive analysis using predefined categories and developed an evidence map of randomized controlled trials (RCTs). We found 282 RCTs examining SMIs for HF patients, comparing two to four interventions, primarily targeting individual patients (97%) globally (34 countries, only 31% from an European country). These interventions involved support techniques such as information sharing (95%) and self-monitoring (62%), often through a mix of in-person and remote sessions (43%). Commonly assessed outcomes included quality of life, hospital admissions, mortality, exercise capacity, and self-efficacy. Few studies have focused on lower socio-economic or minority groups. Nurses (68%) and physicians (30%) were the primary providers, and most studies were at low risk of bias in generating a random sequence for participant allocation; however, the reporting was noticeably unclear of methods used to conceal the allocation process. Our analysis has revealed prevalent support techniques and delivery methods while highlighting methodological challenges. These findings provide valuable insights for researchers, clinicians, and policymakers striving to optimize SMIs for individuals living with HF.

## 1. Introduction

Heart failure (HF) is a prevalent condition with significant global health implications. It is estimated that approximately 26 million individuals worldwide are affected, leading to substantial healthcare costs [1]. Moreover, the prevalence of HF has been on the rise in recent decades [2]. Managing HF presents considerable challenges to healthcare systems worldwide. It involves a combination of medical treatments and lifestyle changes aimed at controlling symptoms and improving quality of life [3]. These interventions include medication, substantial dietary changes, increased physical activity, and regular self-monitoring activities, all of which rely on patients’ willingness to accept and take responsibility for their condition [4,5,6].

Self-management interventions (SMIs) have emerged as a promising approach to enhancing HF outcomes. Although various definitions of SMIs exist [7], they are generally characterized as supportive interventions that healthcare professionals, peers, or non-professionals offer to enhance patients’ abilities and self-assurance in managing their chronic conditions. SMIs for HF may include educational programs, guidance on self-monitoring, lifestyle advice, and assistance with behavior change [8]. However, it is important to note that interventions targeting self-management in HF vary substantially in terms of their objectives, content, and the level and type of support provided. Many patients diagnosed with HF find self-management tasks burdensome, impacting their family, work, and social life and presenting daily challenges that persist throughout their lives [9]. Consequently, it is not surprising that numerous patients struggle to incorporate these self-management tasks into their daily routines [10].

The COMPAR-EU project (https://self-management.eu) investigated and compared the most effective SMIs for adults living with HF, along with diabetes, COPD, and obesity, in Europe [11] (Box 1). A taxonomy of SMIs and a core outcome set (COS) were developed [12]. Further details about the project have been published elsewhere [13]. In the context of the project, a systematic review with network meta-analysis on SMIs for HF was conducted. This study is part of COMPAR-EU and utilizes the taxonomy to describe and summarize existing randomized evidence and to identify knowledge gaps regarding SMIs in adults with HF.

Box 1The COMPAR-EU Project [11].  COMPAR-EU is a multimethod, interdisciplinary project that contributes to bridging the gap between current knowledge and practice of self-management interventions (SMIs). COMPAR-EU aims to identify, compare, and rank the most effective and cost-effective SMIs for adults in Europe living with one of the four high-priority chronic conditions: type 2 diabetes mellitus (T2DM), obesity, chronic obstructive pulmonary disease (COPD), and heart failure (HF).  The project provides support for policymakers, guideline developers, and professionals to make informed decisions on the adoption of the most suitable SMIs through an IT platform featuring decision-making tools adapted to the needs of a wide range of end users (including researchers, patients, and industry).  COMPAR-EU launched in January 2018 and was completed in December 2022, contributing the following outputs:   (i) an externally validated taxonomy composed of 132 components, classified in four domains (intervention characteristics, expected patient (or carer) self-management behaviors, type of outcomes and target population characteristics); (ii) core outcome sets (COS) for each disease, including 16 outcomes for COPD, 16 forHF, 13 for T2DM and 15 for obesity; (iii) extraction and descriptive results for each disease based on 698 studies for diabetes, 252 studies for COPD, 288 studies for HF and 517 studies for obesity; (iv) comparative effectiveness analysis based on a series of pairwise meta-analyses, network meta-analysis (NMAs) and component NMAs (CNMA) for all outcomes across all four diseases; (v) contextual analysis addressing information on equity, acceptability and feasibility; general information on contextual factors on the level of patients, professionals, their interaction and the health care organization for those interested in implementation; (vi) cost-effectiveness conceptual models have been created for each chronic condition, including risk factors or intermediate variables relevant for SMIs and final outcomes; (vii) business plans and a sustainability strategy were developed based on a multi-prong approach including qualitative interviews with managers and clinicians, the focus group with clinical representatives from EU countries, workshops with industry representatives and a hackathon event.  Most of the COMPAR-EU end-products are available on the online COMPAR-EU platform: www.self-management.eu (accessed on 13 September 2023). Watch the introductory video about the decision aids: https://youtu.be/_nqy6s79ZcY (accessed on 13 September 2023)

## 2. Materials and Methods

We constructed an evidence map of randomized controlled trials (RCTs) [14,15]. To ensure transparency and adherence to established guidelines, we followed the Preferred Reporting Items for Systematic Reviews (PRISMA) and the extension for scoping reviews (PRISMA-ScR) [16,17]. Prior to data extraction, we established the review methodology, and the protocol was registered in PROSPERO (CRD42020155441) [18].

### 2.1. Search

To find published systematic reviews on SMIs, we first searched the database of a previous European project (PRO-STEP) between 2000 and 2015. Then, we updated this dataset through new searches in MEDLINE, CINAHL, Embase, Cochrane, and PsycINFO from 2010 to 2018 (Appendix A). This search string was repeated with terms related to comorbidity, gender, minority groups, socio-economic status (SES), and health literacy to adjust for diversity. The final search was developed, run, and adapted to the requirements of each database by an expert librarian.

### 2.2. Eligibility Assessment

As part of the screening process, we considered studies involving SMIs for individuals aged 18 years and older with HF, as well as their caregivers. Our emphasis was placed on RCTs that incorporated one or more outcomes outlined in the core outcome set (COS) for HF, which was established during earlier phases of the COMPAR-EU project [12]. A full description of the inclusion and exclusion criteria can be found in Appendix A.

We considered SMIs as actions systematically provided by healthcare staff or other patients or laypersons to increase patients’ skills and confidence in managing their chronic conditions. SMI aims to equip patients (and their informal caregivers whenever appropriate) to actively participate in the management of their condition [19].

Pairs of two independent reviewers conducted the initial screening of retrieved study titles and abstracts. Prior to screening, all researchers underwent a calibration exercise to align with a gold standard established by experienced supervisors.

Each reviewer assessed 20 studies for inclusion based on title and abstract, with any uncertainties resolved through discussions with an experienced third researcher. A minimum of 80% agreement with the gold standard was required for calibration before proceeding to screen the remaining titles and abstracts. The screening of titles and abstracts of the retrieved studies was carried out using Covidence (www.covidence.org).

For full-text screening, a similar calibration exercise was conducted for 10 full-text papers. The final determination of eligibility for each included full-text study was made by two independent researchers and confirmed by a supervisor through consensus.

### 2.3. Data Extraction

We utilized a standardized form integrated into an online platform developed in the project’s earlier stages. One reviewer collected pertinent data from eligible studies, with a second reviewer ensuring accuracy. The data extraction form adhered to the COMPAR-EU taxonomy structure for SMIs, as depicted previously.

The form encompassed patient characteristics, including comorbidities, gender, and SES variables (e.g., health literacy), SMI support techniques, expected self-management behaviors, study outcomes, study design information, and risk of bias assessment. Multiple publications reporting separate aspects of a study were considered to be one study. Measures and tools for outcomes were included in the platform, with additional ones added as needed.

The taxonomy of SMIs provides a comprehensive framework that encompasses various support techniques and anticipated self-management behaviors [12]. Twelve self-management methods were included in the support techniques: sharing information, skills training, self-monitoring, prompt use, goal setting, problem-solving, coaching, emotional management, social support, shared decision-making, service use, and provision of equipment. In addition, the core outcome set established by the COMPAR-EU project encompasses essential aspects such as empowerment, adherence to self-management behaviors, clinical outcomes, quality of life for patients and informal caregivers, perceptions of care satisfaction, and cost considerations [12].

Given the extensive gaps in precise intensity data, we evaluated each study arm to determine whether the cumulative patient-provider contact time amounted to at least 10 h, which was considered to be high intensity. If the intensity of contact was unclear, we assumed low intensity. This analysis combined data from both face-to-face and remote sessions.

Additionally, a risk of bias assessment was conducted using Cochrane’s risk of bias tool, with a second reviewer validating the judgments. When insufficient information was available, study authors were contacted for clarification or additional data. We categorized the risk of bias as low, high, or unclear in each of the five domains of the Cochrane “Risk of Bias” tool for all included studies.

### 2.4. Data Analysis

Descriptive analysis was used to synthesize the study’s findings.

## 3. Results

### 3.1. Search Results

Our search yielded a total of 6093 citations, comprising primary studies from various sources: PRO-STEP (*n* = 966), PubMed (*n* = 2139), Embase (*n* = 935), CINAH (*n* = 1809), PsycINFO (*n* = 23), and Cochrane (*n* = 221). After the initial title and abstract screening, 690 citations progressed to full-text screening. Ultimately, 282 of these studies were included in the descriptive analysis. The primary causes for exclusion during the extraction phase (*n* = 60) were no SMI (*n* = 23), no RCT (*n* = 16), no outcomes from COS (*n* = 15), or wrong population (*n* = 6). The flowchart (Figure 1) describes the process in more detail.

### 3.2. Characteristics of Included RCTs

The 282 studies were conducted in 34 different countries; 87 were conducted in Europe (30.8%). By far, most of the studies were conducted in the United States (*n* = 112, 39.7%), followed by Sweden (6.0%) and the United Kingdom (5.3%). The total number of participants varied widely, ranging from 1482 (for adherence) to 30,106 (for mortality). Most studies were implemented on an individual patient level (96.8%), with a smaller percentage focused on the population level. Of the studies, 55.7% were concerned with single-center studies, and 41.5% were conducted in multiple centers. Studies started between 1994 and 2017, with 29.1% (*n* = 82) starting between 2010 and 2019; 202 studies reported no private for-profit funding (71.6%), 34 studies reported private for-profit funding (12.1%), and 46 studies funding was unclear or not reported (16.3%).

Across all studies, most of them focused on patients (84.0%), with only four studies for caregivers and 40 aimed at both patients and caregivers (Table 1).

In almost 90.0% of the studies, an SMI was compared to usual care; In 65.3% of the cases, usual care consisted of regular clinical visits and a form of education. In one third of the studies (34.7%), usual care was a little more intensive (“usual care plus”), including also other self-management support techniques such as coaching. The number of intervention arms ranged from two to four, yet most of the studies (93.3%) included two arms. In 23.0% of the intervention arms, the intervention was tailored. For instance, this involved simplifying existing interventions in response to respondents with low health literacy or adjusting interventions due to recognized gender differences between men and women.

### 3.3. Characteristics of the Participants

Participants were most often male (median 67%, IQR 59–74%); the median age was 67 years old (IQR 62–73 years), and the median time since diagnosis was 4.5 years across all studies. Although most studies involved general samples of HF patients, others employed more targeted inclusion criteria. Specifically, six studies (2.1%) concentrated on populations with a low SES. Only 12 studies (4.2%) reported specifically on populations from minority groups. There were only three studies that provided information on health literacy levels.

Regarding disease-related variables, when more than one measure of disease severity was given, NYHA Function Classification was the preferred measure, followed by Left Ventricle Ejection Fraction (LVEF). Fifty studies (17.8%) reported on the time since diagnosis of HF patients, and 254 studies (90.1%) provided information on the severity of HF, with 74% providing information on NYHA at baseline (median 2.7, IQR 2.3–3.0) and 22% providing information on LVEF (median 32.7, IQR 27.9–37.5). In 62 studies (21.9%), information on comorbidities was reported. The number of comorbidities ranged from 1 to 8; in 42 of the 62 studies, the number of comorbidities was not specified. Hypertension, diabetes, depression, and COPD were the most common comorbidities encountered.

### 3.4. Self-Management Support Techniques

Table 2 shows the mean number and frequency of each possible support technique for the usual care arms (*n* = 250) and intervention arms (*n* = 336). Figure 2 displays a matrix illustrating the frequency with which specific SM support techniques are combined across various studies.

Within usual care, 45.6% of the studies reported none of the 12 support techniques. In those usual care arms that mentioned one or more techniques, sharing information (54.0%) was the mentioned most frequent often (*n* = 135). Of those studies using information, 71 studies used previously designed materials like leaflets, websites, or other media. The intervention arms employed a range of one to ten self-management support techniques (median 4, IQR 3, 6). Commonly provided techniques included sharing information (94.6%), self-monitoring (61.9%), skills training (49.7%), goal setting (35.4%), and provision of equipment (35.1%). Shared decision-making was mentioned the least in only six studies. Looking at the matrix (Figure 2), it appears that in the 282 studies, especially combinations of sharing information with self-monitoring and/or goal setting were offered frequently (>100 studies). The same counts for self-monitoring in combination with equipment provision and/or goal setting (~90 studies). Skills training was offered frequently in combination with sharing information and self-monitoring (>100 studies).

#### 3.4.1. Support Delivery Methods

In 47.6% of the descriptions within the usual care arm, information about the delivery method was missing. Among those who provided information, the primary form of support was clinical visits (35.6%).

In the intervention arms, 43.2% of the arms used a combination of clinical visits, support sessions, and self-guided sessions; 33.3% used only support sessions, and 14.6% used only clinical visits. In most intervention arms (65.2%), the SMI was delivered synchronously, implying simultaneous communication between patients and providers. Additionally, 30.1% used a combination of synchronous and asynchronous communication.

This pattern is also evident in the mode of delivery. A little more than half of the intervention arms (53.0%) used a combination of face-to-face and remote support sessions, for example, by phone or computer, whereas in usual care, support seems to be given mainly face-to-face during clinical visits (33.2%). Interventions were delivered most of the time to individual patients (80%).

#### 3.4.2. Location

Most intervention arms took place in a single location (56.5%), with over 40% taking place in multiple locations. Home care (57.1%) and outpatient care (35.6%) were far more often used as location; SMIs for HF were rarely (>20%) administered in hospitals, long-term care facilities, community settings, work environments, or virtually.

#### 3.4.3. Type of Provider

In most intervention arms (49.4%), only one provider was engaged, with the most common roles being nurses, physiotherapists, or physicians. Peers, laypersons, and social workers played limited roles in HF interventions, and interventions that did not mention any providers were exclusively remote and self-guided. Refer to Figure 3 for a comprehensive overview of provider combinations. In the case that a nurse, who is the most frequently involved provider (229 studies), works together with someone, this is most often with a physician (75 studies), health service (25 studies), or nutritionist (24 studies).

#### 3.4.4. Duration and Intensity

There were 259 intervention arms (77.1%) that involved face-to-face contact. This number encompasses both interventions solely reliant on face-to-face interactions and those employing a combination of face-to-face and remote methods. Out of these intervention arms, we were able to calculate the total time dedicated in minutes for 99 cases (38.2%). The median duration spent on face-to-face contact was 180 min (with an interquartile range of 60–480 min). Notably, this time allocation was significantly greater than that observed in the usual care, where the median time spent was 55 min (with an interquartile range of 47–186 min).

Among usual care arms, 99.6% were categorized as low intensity. In contrast, within the intervention arms, 22.0% reported contact time of at least 10 h.

Among the 336 intervention arms, we calculated the total time allocated to remote interventions in only 28 instances. From these interventions, the median time spent (98 min) was similar to the median time for remote interventions within the usual care group.

### 3.5. Expected Self-Management Behaviors

Within usual care, the expected behaviors of patients were often not specified (68.0%). In those usual care arms that did mention the expected behaviors they focus on, the number of behaviors varied between one and nine and most often concerned behavior in relation to physical activity (14.8%), healthy eating (16.4%), medication use (16.0%) and self-monitoring (12.8%). The range of expected behaviors mentioned in the intervention arms varied from one to nine (median 4, IQR 2–6). Like in usual care, expected behaviors most often include self-monitoring (70.8%), being physically active (54.2%), healthy eating (53.9%), medication use (60.1%), and the early recognition of symptoms (44.0%); behaviors in relation to work and social roles, and the interaction with health care, healthy sleeping, and alcohol use are seldom mentioned. Regarding combinations, it appears that in the 336 intervention arms, especially the combinations of self-monitoring + medication use (*n* = 153), healthy eating + early recognition of symptoms (*n* = 104), healthy eating + physical activity (*n* = 125), and physical activity + medication use (*n* = 120) goes together.

### 3.6. Outcomes

Regarding outcomes, the COS for HF consisted of 19 outcomes. Table 3 demonstrates significant heterogeneity in the frequency at which outcomes were considered. The most frequently measured outcomes were quality of life (57.1%), hospital admissions (56.7%), mortality (37.2%), exercise capacity (25.2%), and self-efficacy (24.1%). Other outcomes like adherence, body weight, or knowledge were less frequently studied. The same outcomes were measured in many different ways. Appendix A gives an overview of outcomes and tools.

### 3.7. Risk of Bias

In terms of selection bias, most of these studies demonstrated a low risk of bias in generating a random sequence for participant allocation (Figure 4). However, there was a notable lack of clarity regarding the reporting of methods used to conceal the allocation process.

The main methodological limitation observed across the included studies pertained to the absence of blinding in the interventions. Only a small number of studies incorporated strategies to conceal the active intervention from both participants and care personnel or used a “sham” intervention to mitigate the participants´ awareness of their assigned study arm. This limitation affected the evaluation of subjective outcomes like quality of life, as well as objective outcomes that could potentially be influenced by assessors, such as blood pressure. However, it was observed that objective outcomes derived from laboratory tests or based on clearly observable events (e.g., mortality, hospitalization) were less susceptible to the lack of blinding.

Approximately 36% of the studies also experienced a notable level of drop-out during the follow-up period, which raised concerns about a high risk of bias due to attrition in these cases. Evaluating the risk of selective reporting proved more challenging because only a limited number of studies presented their protocols prior to publishing results. Nevertheless, most studies were deemed to have a low risk of bias in this aspect.

## 4. Discussion

### 4.1. Main Findings

Our study offers a systematic overview of RCTs focusing on SMIs designed for adults living with HF, specifically targeting outcomes relevant to patients. We described SMIs from studies published between 2010 and 2018, detailing their contents and primary characteristics. This includes the self-management support techniques incorporated, targeted behaviors, delivery methods, and the extent to which outcomes important to patients were assessed.

We identified a total of 282 RCTs that investigated the effectiveness of SMIs for patients with HF. These studies compared two and four different interventions, with the majority primarily targeting individual patients (97%) across a global landscape that encompassed 34 different countries, with only 31% of these studies conducted in European countries. The SMIs in these trials predominantly involved support techniques, such as information sharing (95%) and self-monitoring (62%), often delivered through a combination of in-person and remote sessions (43%).

The most commonly assessed outcomes in these trials included measures of quality of life, rates of hospital admissions, mortality, exercise capacity, and self-efficacy. However, it is worth noting that few of the included studies specifically focused on populations with lower socio-economic status or minority groups. Nurses played a central role as the primary providers of these interventions in 68% of the studies, followed by physicians in 30% of the cases. Notably, most of the included studies were deemed to be at low risk of bias, which enhances the reliability of their findings and underscores the robustness of the evidence generated from these trials.

### 4.2. Research in Context

To our knowledge, there is currently no systematic evidence on the composition and structure of SMIs for HF, similar to the comprehensive approach we applied in this study. Although various systematic reviews have previously summarized or examined specific components of SMIs in isolation [9,20,21,22], such as self-management educational programs, lifestyle modifications, education, and patient empowerment, none have taken the integrated approach we used. We employed evidence mapping and adhered to a taxonomy to offer a comprehensive overview of SMIs. In terms of intervention reporting, our investigation revealed a significant absence of information regarding the precise design, intensity, and operationalization of outcomes. These findings are consistent with similar studies conducted within the COMPAR-EU initiative for chronic conditions like T2DM, COPD, and obesity, which also identified deficiencies in reporting mode of delivery, intensity, location, and provider involvement, along with significant heterogeneity in the reporting of these variables across studies [23,24].

Our study revealed that more than 20% of SMIs in the context of HF were customized to suit the needs of the study population. This customization involved adjusting both the content and the delivery method of the intervention. This percentage is significantly higher than the less than 5% of tailored studies observed in the context of SMIs for diabetes and obesity, highlighting the advanced state of research in self-management for HF when compared to other chronic conditions. Notably, this 20% figure is similar to what was found for T2DM interventions, indicating a parallel trend in tailoring approaches for both HF and diabetes. Collectively, our findings suggest that SMIs in the context of HF should integrate personalized, gender-specific, and eHealth-oriented strategies to improve patient outcomes and quality of life [20,25,26].

Our COMPAR-EU group conducted a study evaluating the cost-effectiveness models for SMIs in HF alongside other chronic conditions across Germany, Greece, the Netherlands, Spain, and the UK. Our analysis revealed varying “headrooms”, indicating the maximum acceptable price at which an intervention may still be cost-effective, ranging from EUR 0 to EUR 2406 and EUR 218 to EUR 8031 at EUR 20,000 and EUR 50,000 thresholds, respectively. For HF, our findings suggest that SMIs with relatively high “headrooms” may be delivered cost-effectively. This potential, however, largely depends on the actual costs of implementing SMIs in practice. Understanding the duration of SMI effects proved crucial, impacting our assessment. Overall, budgetary impact varied, emphasizing the need for tailored approaches, considering disease prevalence and population size.

### 4.3. Strengths and Limitations

Our study has distinct strengths. First, we conducted a comprehensive search across multiple databases, including PubMed, Embase, CINAHL, PsycINFO, and Cochrane, as well as utilizing previous findings from PRO-STEP, to identify SMIs for HF. Second, our mapping process adhered to a pre-established taxonomy for SMIs, allowing us to evaluate the individual elements or components of the structured health interventions in relation to the outcomes that are significant for the patients. This meticulous approach offers a valuable synopsis of commonly utilized components and uncovers avenues for deeper investigation, streamlining research design and avoiding duplication. Lastly, our utilization of visual aids like figures and tables effectively translates complex information and patterns for better understanding.

Nevertheless, our study has several limitations. Initially, there is the potential for overlooking newly emerged SMIs, such as digital health interventions aiming to enhance the adoption of self-management education and personalized mHealth interventions designed to support medication adherence and encourage lifestyle changes. This is due to the evidence map’s data coverage being up until 2018. Additionally, while we provided a comprehensive description and visualization of the existing evidence and gaps within the SMIs field, we did not provide an operational search engine for users. However, to aid navigation through current evidence, we have made all the RCTs available on the COMPAR-EU platform.

## 5. Conclusions

In conclusion, this comprehensive analysis of 282 RCTs on SMIs for HF sheds light on the multifaceted landscape of interventions aimed at enhancing patient outcomes. Although common support techniques and delivery methods have been identified, challenges in blinding and outcome measurement heterogeneity highlight areas for improvement. It is worth noting that few of the included studies specifically focused on populations with lower SES or minority groups. These findings provide valuable insights for researchers, clinicians, and policymakers striving to optimize self-management strategies for individuals living with HF.

## Figures and Tables

**Figure 1 healthcare-12-00302-f001:**
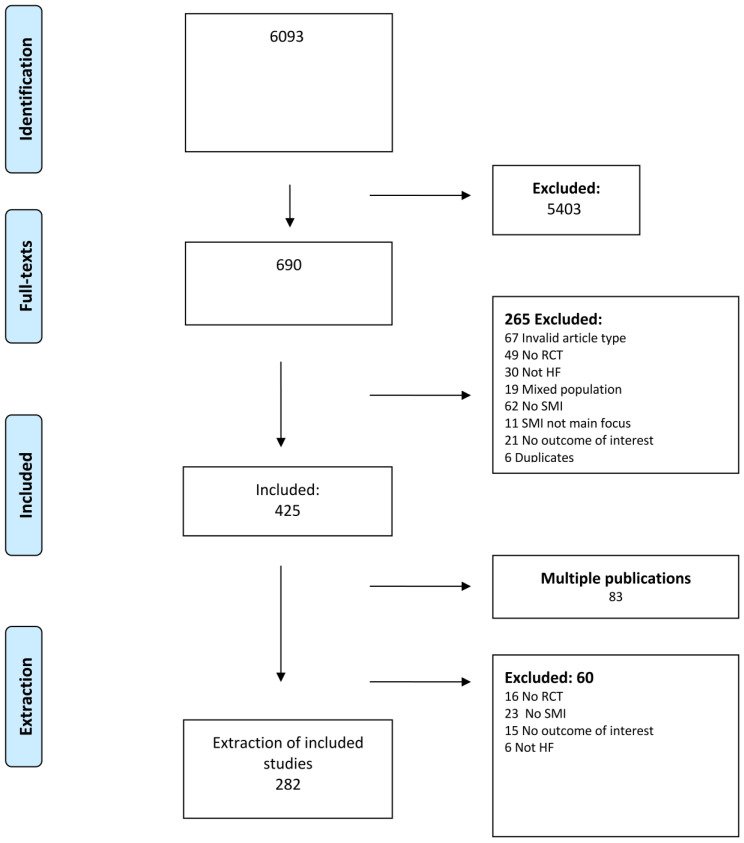
Preferred Reporting Items for Systematic Reviews and Meta-Analyses (PRISMA) flow chart.

**Figure 2 healthcare-12-00302-f002:**
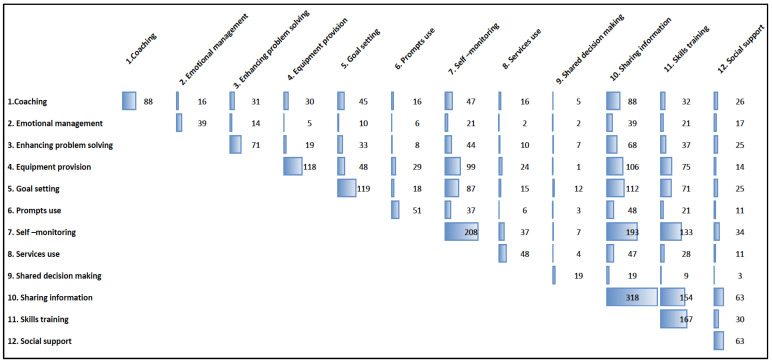
Frequency in which support techniques are combined across intervention arms (*n* = 336).

**Figure 3 healthcare-12-00302-f003:**
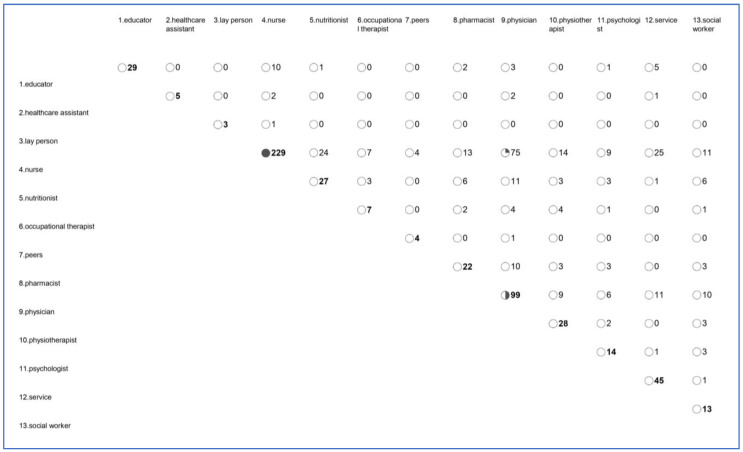
Providers’ combinationss frequency across intervention arms (*n* = 336).

**Figure 4 healthcare-12-00302-f004:**
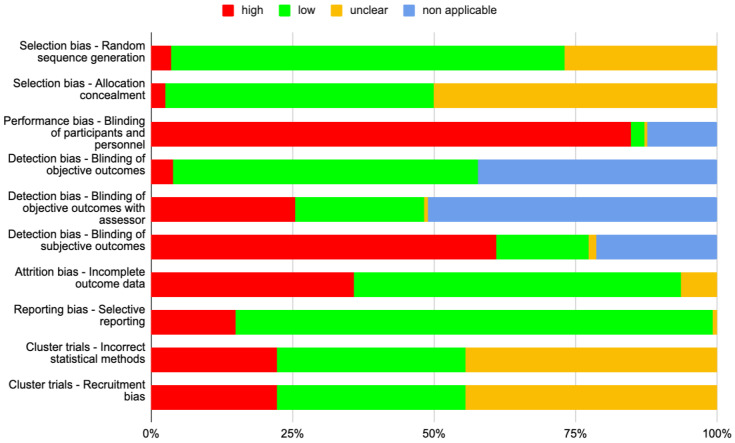
Risk of bias (*n* = 282).

**Table 1 healthcare-12-00302-t001:** Characteristics of included RCTs.

Variables		*n* (%)
Type of population (*n* = 282)	patient	237 (84.0)
	caregiver	4 (1.4)
	mixed	40 (14.2)
	unknown	1 (0.4)
Type of comparison (*n* = 282)	head-to-head	32 (11.4)
	intervention(s) vs. UC	248 (88.6)
Type of usual care * (*n* = 248)	UC	162 (65.3)
	UCP	86 (34.7)
Number of intervention arms (*n* = 282)	2	263 (93.3)
	3	16 (5.7)
	4	3 (1.1)
Number of arms (head-to-head) (*n* = 32)	2	30 (94.0)
	4	2 (6.0)
Number of arms (intervention vs. UC) (*n* = 248)	2	231 (93.1)
	3	16 (6.5)
	4	1 (0.4)

UC: usual care; UCP: usual care plus. * A UC arm was typically defined by study authors and generally included standard of care treatment and visits plus an educational component. When it included more components, even any kind of self-management support technique (for example, counseling), but it was still defined as usual care by the study authors, we called this UCP.

**Table 2 healthcare-12-00302-t002:** HF support techniques used within usual care and intervention arms (*n* = 282) *.

		Usual Care *n* = 250	Intervention Arms *n* = 336
Number of support techniques	0	114 (45.6)	0
	1	50 (20.0)	6 (1.8)
	2	66 (26.4)	40 (11.9)
	3	16 (6.4)	51 (15.2)
	4	2 (0.8)	82 (24.4)
	5	1 (0.4)	60 (17.9)
	6	1 (0.4)	52 (15.5)
	7	0	28 (8.3)
	8	0	8 (2.4)
	9	0	6 (1.8)
	10	0	3 (0.9)
Number of support techniques		1 (0, 2)	4 (3, 6)
Coaching		2 (0.8)	88 (26.2)
Emotional management		0	39 (11.6)
Enhancing problem-solving		0	71 (21.1)
Equipment provision		13 (5.2)	118 (35.1)
Goal setting		0	119 (35.4)
Previously designed materials		71 (28.4)	214 (63.7)
Prompts use		0	51 (15.2)
Self-monitoring		16 (6.4)	208 (61.9)
Services use		4 (1.6)	48 (14.3)
Shared decision-making		0	19 (5.7)
Sharing information		135 (54.0)	318 (94.6)
Skills training		5 (2.0)	167 (49.7)
Social support		3 (1.2)	63 (18.8)
No specific SM support		101 (40.4)	0

* Data are presented as median (IQR) for continuous measures and *n* (%) for categorical measures.

**Table 3 healthcare-12-00302-t003:** Frequency of outcomes from the Core Outcomes Set (COS).

	*n* (%)
Quality of life	161 (57.1)
Hospital admissions	160 (56.7)
Mortality	105 (37.2)
Exercise capacity (including effort test)	71 (25.2)
Self-efficacy	68 (24.1)
Adherence to medication or other treatment	37 (13.2)
Knowledge	37 (13.1)
Breathlessness (dyspnea)	15 (5.3)
Physical Activities	14 (5.0)
Body Weight (anagement)	13 (4.7)
Adherence to diet as agreed (including salt and water)	7 (2.5)
Caregiver quality of life	6 (2.1)
Self-monitoring	6 (2.1)
Value for money of the self-management intervention	5 (1.8)
Perception of health care professional relationship and communication	4 (1.4)
Patient activation	3 (1.1)
Swelling (including leg and abdominal edema)	2 (0.7)
Health literacy	1 (0.4)
Participation and decision-making	1 (0.4)

## Data Availability

Data (transcripts) from the COMPAR-EU project have been deposited according to the data management plan at NIVEL. The qualitative datasets used and/or analyzed during the current study are available from the corresponding author upon reasonable request, provided they do not identify interviewees.

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
