# Peer review of "Effectiveness and Cost-Effectiveness of Self-Management Interventions for Adults Living with Heart Failure to Improve Patient-Important Outcomes: An Evidence Map of Randomized Controlled Trials"

_healthcare, 2024, doi:10.3390/healthcare12030302_

Round 1
Reviewer 1 Report
Comments and Suggestions for Authors
The authors present their study titled "Comparing the effectiveness and cost-effectiveness of self-management interventions for adults living with heart failure to improve patient-important outcomes: An evidence map". It is a systematic review describing the types of self-management interventions in heart failure patients by collecting data from multiple randomized trials during 2010-2018. They end up including 282 studies and describing 12 support techniques. In particular, sharing information, self-monitoring, skill training, goal setting and provision of equipment were the more frequent ones. They also describe, among other data, the location where the services were offered (more often homecare and outpatient care), the type of provider (more often a nurse) and the studied outcomes of those trials (more frequently quality of life, hospital admissions and mortality).
The article is very well written, following a systematic approach according to PRISMA recommendations while also providing comprehensive results on a subject that is not commonly reviewed. They also adequately describe the study limitations, especially the lack of telemedicine methods due to the studied period (pre-2018 era).
One thing that I would like to see - if there are available data in the included studies - is a brief comment on how many studies had cardiopulmonary exercise testing parameters as indexes of heart failure severity or outcomes (in particular peakVO2) given their important prognostic value.
Author Response
Thank you for bringing this to our attention. We agree with your comment. Upon re-evaluation of the included studies, it's important to note that severity of illness was evaluated only according NYHA scale and as an outcome, when reported, it was pooled with other measure/tools as exercise capacity. Regrettably, we were unable to retrieve this particular information on peakVO2. [No changes have been made to the manuscript]
Reviewer 2 Report
Comments and Suggestions for Authors
Self-management interventions (SMIs) may improve heart failure (HF) outcomes and address illness management issues. The purpose of this study was to conduct a descriptive analysis using predetermined categories and to create an evidence map of randomized controlled trials (RCTs) inside the COMPAR-EU project by scanning the databases MEDLINE, CINAHL, Embase, Cochrane, and PsycINFO from 2010 to 2018. 31. Actually, the current proposal is interesting and well-written. Therefore, I recommend that the current study be published after minor revisions as follows:
1- Could the authors discuss the incremental ratio of cost-effectiveness analysis?
2- Please discuss the future directions for SMIs
3- Please add a diagrammatic figure to propose the possible mechanistic pathway for these findings
Author Response
Thank you for your valuable suggestions. We concur with your recommendations and have taken the following actions in response:
1- Regarding the incremental ratio of cost-effectiveness analysis, we have included a detailed discussion in the revised manuscript. In the "Discussion" section on page 12, paragraph 3, lines 360-369, we have provided the results of the cost-effectiveness analysis performed by our group, outlining its implications and significance in the context of the study's findings.
The current sentence reads as follows:
“Our COMPAR-EU group conducted a study evaluating the cost-effectiveness models for SMIs in HF alongside other chronic conditions across Germany, Greece, the Netherlands, Spain, and the UK. Our analysis revealed varying 'headrooms,' indicating the maximum acceptable price at which an intervention may still be cost-effective, ranging from €0 to €2,406 and €218 to €8,031 at €20,000 and €50,000 thresholds, respectively. For HF, our findings suggest that SMIs with relatively high 'headrooms', maybe delivered in a cost-effective manner. This potential, however, largely depends on the actual costs of implementing SMIs in practice. Understanding the duration of SMI effects proved crucial, impacting our assessment. Overall, budgetary impact varied, emphasizing the need for tailored approaches, considering disease prevalence and population size.”
2- To address the future directions for Self-Management Interventions (SMIs), we have incorporated a dedicated section in the "Discussion" on page 12, starting from paragraph 2, lines 356-359. This segment now outlines potential avenues for further research, highlighting areas that necessitate deeper exploration and potential advancements in SMIs for heart failure management. Additionally, we have supplemented this section with relevant references to strengthen the discussion and support the proposed future research directions.
The current sentence reads as follows:
"Collectively, our findings, suggest that SMIs in the context of HF should integrate personalized, gender-specific, and eHealth-oriented strategies to improve patient outcomes and quality of life [25-27]."
3- Regarding your suggestion about including a diagram illustrating possible mechanistic pathways, we'd like to clarify that our study was primarily descriptive and exploratory in nature. Our main focus was to conduct a descriptive analysis and create an evidence map of RCT within the COMPAR-EU project, rather than delve into specific mechanistic pathways behind these interventions. Given the descriptive nature of our study, which aimed to summarize available evidence and categorize interventions, we did not explore detailed mechanistic pathways in our findings. [No changes have been made to the manuscript]
Reviewer 3 Report
Comments and Suggestions for Authors
In the manuscript healthcare-2733421 entitled “Comparing the effectiveness and cost-effectiveness of self-management interventions for adults living with heart failure to improve patient-important outcomes: An evidence map”, Dr. Santero and colleagues overviewed the designs of studies published from 2010 to 2018 which targeted the efficacy of self-management interventions (SMIs) for patients with heart failure. They reviewed 282 randomized control trials and summarized the trends in types of the interventions, targeted outcomes and providers. Looking back these accumulated evidences, the authors raised the problems in the previous studies and discuss to propose necessary viewpoints for futural studies.
Although the study has some interest, there seems several concerns to be dissolved as following.
1. Although the title of the manuscript was named as "Comparing the effectiveness and cost-effectiveness of self-management interventions for adults living with heart failure to improve patient-important outcomes: An evidence map", effectiveness and cost-effectiveness were not centered in this review. Because the authors focused on the trend of methods and designs of previous studies, it should be reflected in the title as appropriate. In addition, "Patient-Important Outcomes" seems difficult to be understood.
2. The authors described sex, age, NYHA classification, LVEF and numbers of comorbidities for the characteristics of participants in the text. To understand the targeted patients more clearly, some additional data should be presented. eg. acute or chronic heart failure, NT-proBNP concentration or existence of atrial fibrillation.
3. While the reviewer understand that the authors focused on the designs of the previous studies in the current investigation, the results of the studies are also hoped to be presented.
Comments on the Quality of English LanguageSome typos should be corrected properly.
eg. Line 217
Shared decision-making decision making was mentioned the least, in only six studies.
Author Response
Thank you for your valuable suggestions. We concur with your recommendations and have taken the following actions in response:
1- We appreciate your perspective and the suggestion to refine the title to better reflect the content of our study. We understand and acknowledge that our primary objective was not centred on exploring methodological trends. Our focus was indeed to create an evidence map summarizing available evidence on self-management interventions for heart failure, rather than emphasizing methodological trends. Regarding the term "Patient-Important Outcomes," it's a widely used term in the field, as evidenced in our previous articles, published in the same journal (https://pubmed.ncbi.nlm.nih.gov/38132046/)
Based on your feedback, here's a revised title for the manuscript:
"Effectiveness and cost-effectiveness of self-management interventions for adults living with heart failure to improve patient-important outcomes: An evidence map of randomized controlled trials"
2- Thank you for bringing this to our attention. We agree with your comment. Regrettably, we were unable to retrieve this particular information because it was not extracted. [No changes have been made to the manuscript]
3- We regret any misunderstanding caused by the lack of detailed results, as it is beyond the primary scope of our study. [No changes have been made to the manuscript]
Thank you for flagging the typo in line 217. We corrected it promptly and conducted a comprehensive proofread for language accuracy throughout the manuscript.